# Consequences of Transition Treatments on Fertility and Associated Metabolic Status for Dairy Cows in Early Lactation

**DOI:** 10.3390/ani10061100

**Published:** 2020-06-25

**Authors:** Junnan Ma, Renny J. van Hoeij, Rupert M. Bruckmaier, Akke Kok, Theo J. G. M. Lam, Bas Kemp, Ariette T. M. van Knegsel

**Affiliations:** 1Department of Animal Sciences, Adaptation Physiology Group, Wageningen University, P.O. Box 338, 6700 AH Wageningen, The Netherlands; junnan.ma@wur.nl (J.M.); rennyvanhoeij@gmail.com (R.J.v.H.); akke.kok@wur.nl (A.K.); bas.kemp@wur.nl (B.K.); 2Veterinary Physiology, Vetsuisse Faculty, University of Bern, CH-3001 Bern, Switzerland; rupert.bruckmaier@vetsuisse.unibe.ch; 3Department Farm Animal Health, Utrecht University, P.O. Box 80151, 3508 TD Utrecht, The Netherlands; T.Lam@gddiergezondheid.nl; 4GD Animal Health, P.O. Box 9, 7400 AA Deventer, The Netherlands

**Keywords:** dry period length, dietary energy level, ovarian activity, metabolic status

## Abstract

**Simple Summary:**

Shortening or omitting the dry period improves energy balance and metabolic status, but reduces milk production and increases the risk of body fattening of cows in the subsequent lactation. Reducing the postpartum dietary energy level in order to match the lower milk yield after 0-d dry period could prevent body fattening. Earlier, reducing postpartum dietary energy level for cows after 0-d dry period reduced days open in the subsequent lactation, which may indicate improved underlying fertility. This study investigated effects of reducing dietary energy level from week 4 postpartum onwards for cows after 0-d dry period on fertility variables and associated metabolic status. Reducing the postpartum dietary energy level in cows with 0-d dry period reduced the interval from calving to onset of luteal activity in cows of parity ≥ 3, compared with a standard dietary energy level or a 30-d dry period. Fewer days open was related to fewer services per conception, fewer days to onset of luteal activity, higher percentage of ovarian cycles of normal length (18–24 d), and improved energy balance in weeks 1–7 of lactation. In conclusion, reducing a postpartum dietary energy level to match lower milk yield after 0-d dry period improved fertility in cows of parity ≥ 3, but not in cows of parity 2.

**Abstract:**

This study aimed to (1) investigate effects of reducing postpartum dietary energy level for cows after a 0-d dry period (DP) on resumption of ovarian cyclicity and reproductive performance, (2) relate days open with other reproductive measures, and (3) relate onset of luteal activity (OLA) and days open with metabolic status in early lactation. Holstein-Friesian dairy cows were randomly assigned to 1 of 3 transition treatments: no DP and low postpartum dietary energy level from 22 days in milk( DIM )onwards (0-d DP (LOW)) (*n* = 42), no DP and standard postpartum dietary energy level (0-d DP (STD)) (*n* = 43), and a short DP and standard postpartum dietary energy level (30-d DP (STD)) (*n* = 43). Milk progesterone concentration was determined three times per week until 100 DIM. Plasma metabolite and hormone concentrations were measured weekly until week 7 postpartum. Reducing postpartum dietary energy level in older cows (parity ≥ 3) after no DP and 22 DIM did not affect milk production but prevented a positive energy balance and shortened the interval from calving to OLA. In addition, services per pregnancy and days open were reduced in cows of parity ≥ 3 on 0-d DP (LOW), compared with cows of parity ≥ 3 with 0-d DP (STD), but not in cows of parity 2.

## 1. Introduction

Shortening or omitting the dry period (DP) length were reported to improve fertility, indicated by an earlier onset of first ovulation postpartum [1] and overall improved resumption of ovarian cyclicity [1,2,3]. Consistent with the concept that earlier first ovulation may improve reproductive performance, omitting the DP was associated with an increased percentage of cows pregnant to first artificial insemination (AI) (55%, 26%, and 20% for 0-d, 28-d, and 56-d DP, respectively) and decreased days open (94, 121, and 145 d for 0-d, 28-d, and 56-d DP, respectively) [4]. Moreover, in another study, first-service pregnancy rate was greater in multiparous cows with a 35-d DP compared with cows with a longer DP [5]. In some other studies, however, no effects of DP length on pregnancy rate and days open were found after a short DP [6,7] or an omitted DP [7]. In all these studies, a postpartum dietary energy level was not adjusted to the lower milk yield for cows with 0-d DP or a short DP, compared with cows with a conventional DP.

Unfavorable consequences of omitting the DP can be a reduction in milk yield and fattening of cows in the subsequent lactation [8]. Adjusting dietary energy level in early lactation could be a strategy to limit these negative consequences [9]. Earlier, we reported that reducing postpartum dietary energy level for cows with 0-d DP did not affect milk yield or milk composition, but resulted in a less positive energy balance (EB) and less body weight gain in the subsequent lactation, compared with cows fed postpartum a standard energy level after no DP [9]. Moreover, cows with no DP and low postpartum dietary energy level (0-d DP (LOW)) had less days open than cows with no or a short DP and standard postpartum dietary energy level (0-d DP (STD) or 30-d DP (STD)). Days open is influenced by both the interval between calving to insemination and the success of an insemination [10]. Resumption of ovarian cyclicity, which is a combined trait consisting of onset of ovarian activity and regularity of the ovarian cycles, is related with the timing of the first insemination and success of insemination [11]. It can be hypothesized that the reduced days open for cows with 0-d DP (LOW), as we saw earlier [9], is related with an improved resumption of ovarian cyclicity or an increased conception rate after insemination or both.

Improvement in reproductive performance related with a reduction in DP length can be associated with an improved EB and metabolic status [1,3,4]. Concentrations of plasma glucose, IGF-I, and insulin were greater, but milk yield, plasma concentrations of non-esterified fatty acids (NEFA), and liver tri-acyl glycerides (TAG) were lower for cows after a 0-d DP than for cows after a 30-d or 60-d DP [3]. These endocrine and metabolic factors were associated with ovarian activity postpartum [12,13,14]. Elevated concentrations of NEFA and β-hydroxybutyrate (BHB) in plasma reduced oocyte and blastocyst quality in vitro [15,16] and were associated with a later onset of luteal activity (OLA) in vivo [3,17]. In particular, IGF-1 and insulin are key factors for the ovarian function because they both stimulate estradiol-17β production in granulosa cells in vitro [18,19] and proliferation of follicular cells [20,21] in vitro. Contradicting, high plasma insulin concentration had negative effects on oocyte maturation in vitro [15] and insulin stimulating diets are possibly beneficial to establish ovarian activity postpartum, but not to establish pregnancy [16]. Therefore, it can be hypothesized that for reproductive performance, with days open being the net result of resumption of ovarian cyclicity and pregnancy rates, an optimal energy balance and metabolic status would be essential rather than a maximal EB or insulin concentration.

Earlier, we reported consequences of reducing postpartum dietary energy level for cows after no DP on energy balance, metabolic status and days open, as compared with cows after 0-d DP or a short DP with standard dietary energy level [9]. The overall aim of the current study was to unravel the effect of reducing postpartum dietary energy level on days open for cows after 0-d DP. The first objective of the current study was to investigate the effects of reducing postpartum dietary energy level for cows after 0-d DP on resumption of ovarian cyclicity, and reproductive performance, as compared with cows after 0-dDP or 30-d DP with standard postpartum dietary energy level. The second objective of this study was to relate days open with other reproductive measures. The third objective of this study was to relate OLA and days open with metabolic status in early lactation.

## 2. Materials and Methods

### 2.1. Animals and Housing

The Institutional Animal Care and Use Committee of Wageningen University & Research approved the experimental protocol in compliance with the Dutch law on Animal Experimentation as described earlier (protocol number 2014125; Van Hoeij et al., 2017). The experiment was performed from 27 January 2014 until 9 May 2016. Holstein-Friesian dairy cows (*n* = 128) at Dairy Campus research farm (Lelystad, The Netherlands) were selected based on (1) expected calving interval < 490 days, (2) daily milk yield > 16 kg at 90 days before the expected calving date and (3) no clinical mastitis and SCC < 250,000 cells/mL at 2 final test days before conventional drying off day. Cows pregnant with twins were excluded from the study. Cows were housed in a free stall barn with a slatted floor and cubicles and were milked twice daily (6:00 and 18:00 h). Cows with a 30-d DP were fed a dry cow ration from 7 days before drying off and milked once daily from 4 days before drying off. 0-d dry period cow antibiotics were used in any of the cows in this study.

### 2.2. Experimental Design

The experiment was originally designed to study the effects of DP length, dietary energy level, and mid-lactation ration (glucogenic or lipogenic ration) on milk production, milk composition, EB, plasma metabolites, and lactation persistency during a complete lactation [9]. Cows were blocked by expected calving date, milk yield in previous lactation and parity. We aimed for 40 cows per transition treatment, but because of the long-term characteristic of this experiment we included 2 spare blocks and when cows had to be omitted before calving they were replaced. Within each block of 6 cows, 4 cows were assigned randomly to 0-d dry period (0-d DP) and 2 cows to a short dry period of 30 days (30-d DP). Cows with a 0-d DP were assigned randomly to either a low postpartum dietary energy level (LOW) or a standard postpartum dietary energy level (STD) in early lactation. In the first 3 weeks after calving, dietary energy level was the same for all 3 transition treatments: all cows received 1 kg of concentrate from 10 days before the expected calving date and from 4 DIM concentrate supply increased stepwise for all transition treatments with 0.3 kg/d until 6.7 kg/d at 22 DIM [9]. For cows fed the STD diet, concentrate supply was increased further until 8.5 kg/d at 28 DIM, resulting in a dietary energy level contrast (LOW vs. STD) from 22 DIM onwards (Figure 1). A standard dietary energy level was based on the energy requirement for expected milk yield of 30-d DP cows [21]. All 30-d DP cows were fed an STD. A low dietary energy level was based on the energy requirement for expected milk yield of cows with 0-d DP [21]. Thus, all cows were assigned to one of 3 transition treatments: 0-d DP (LOW) (*n* = 42), 0-d DP (STD) (*n* = 43), 30-d DP (STD) (*n* = 43) (Table 1). From week 8 postpartum onwards, cows received either a glucogenic or lipogenic basal ration. Preliminary analysis showed that mid-lactation ration (glucogenic vs. lipogenic ration) did not affect days open or ovarian cyclicity in the first 100 days in milk (DIM) and was therefore not included in the analysis of this study.

### 2.3. Measurements

#### 2.3.1. Feed Intake and Energy Balance

Daily concentrate intake was recorded by a computerized feeder (Manus VC5, DeLaval, Steenwijk, the Netherlands). Daily ration intake was recorded individually using roughage intake control (RIC) troughs and averaged per week (Insentec, Marknesse, the Netherlands).

Energy balance was calculated as net energy (NE) intake minus NE for maintenance, milk yield and pregnancy per week with Dutch NE system for lactation (VEM system; Van Es et al., 1975). According to the Dutch NE system, the daily requirement for maintenance is 42.4 VEM/kg 0.75 BW per day, milk yield is 442 VEM/kg fat- and protein-corrected milk (FPCM). 1000 VEM is equal to 6.9 MJ NE.

#### 2.3.2. Milk Sampling and Progesterone Assay

Milk samples were collected three times a week (Monday, Wednesday, and Friday) during morning milking from the day of parturition until 100 DIM. Samples were stored at −20 °C until analysis of progesterone (P4) concentration. Milk P4 concentration was measured in whole milk by enzyme immunoassay (Ridgeway Science Ltd., Gloucestershire, UK). The intra-assay and inter-assay coefficients of variation were 4.4% and 16.7%, respectively.

#### 2.3.3. Blood Sampling and Analysis

Blood was collected weekly from calving until 7 weeks postpartum as described earlier [9]. In short, blood samples were collected after the morning milking and between 3 and 1 h before the morning feeding from the coccygeal vein or artery into evacuated EDTA tubes (Vacuette, Greiner BioOne, Kremsmunster, Austria). Concentrations of NEFA and BHB were measured enzymatically using kit no. 994–75409 from Wako Chemicals (Neuss, Germany) and kit no. RB1007 from Randox Laboratories (Ibach, Switzerland,) respectively [22]. The plasma glucose concentration was measured using kit no. 61,269 from BioMerieux (Marcy l’Etoile, France) [22]. The plasma insulin concentration was measured using kit no. PI-12K from EMD Millipore Corporation (Billerica, MA, USA). The plasma IGF-1 concentration was measured using kit no. A15729 from Beckman Coulter (Fullerton, CA, USA).

#### 2.3.4. Definitions of Ovarian Cyclicity

First OLA postpartum was defined within 100 DIM as the moment P4 was 4 ng/mL or higher for 2 or more consecutive milk samples. Ovarian cycle length was defined as the number of days between OLA in one ovarian cycle and the OLA in next ovarian cycle. Regularity of ovarian cyclicity of cows was classified into one of 3 groups according to P4 profile from parturition until 100 DIM [3]:Normal ovarian cycle: cycles with 18–24 days in length.Prolonged ovarian cycle: cycles with more than 24 days in length.Short ovarian cycle: cycles with less than 18 days in length.

The percentages of the different type of cycles per cow were calculated within 100 DIM.

### 2.4. Reproduction Protocol

Cows were inseminated after a voluntary waiting period (VWP) of 50 days until at least 170 DIM. Artificial insemination was performed 12 h after oestrous detection by Lely Qwes-HR Activity Tags (Lely, Maassluis, The Netherlands). Every 4 weeks, pregnancy of cows that were inseminated more than 30 days ago was checked by ultrasound.

### 2.5. Statistical Analysis

Data of 128 cows until 100 DIM were collected, among which 6 cows were entered twice. The numbers of cows per treatment, cows with OLA, cows with complete 1st ovarian cycle and complete 2nd ovarian cycle within 100 DIM are presented in Table 1.

First, regularity of ovarian cyclicity (percentages of normal, prolonged and short cycles per cow within 100 DIM) was analysed with the GLIMMIX procedure of SAS (Version 9.2; SAS institute, Inc., Cary, NC, USA). Fixed effects in the model were transition treatments (0-d DP (LOW), 0-d DP (STD), 30-d DP (STD)) and parity class (2 or ≥3).

Second, days open, days to first OLA, length of first luteal phase, length of first follicular phase and length of the first complete postpartum ovarian cycle, were analysed with MIXED procedure of SAS (Version 9.4; SAS Institute, Inc., Cary, NC, USA). Fixed effects in the model were transition treatments (0-d DP (LOW), 0-d DP (STD), 30-d DP (STD)) and parity class (2 or ≥3).

Third, MIXED procedure of SAS was used to analyse differences in plasma metabolites, hormones and EB among classes for days open or days to OLA. Cows were classified into one of three days open classes with similar animal numbers (short < 80 d, medium 80 to 130 d, long > 130 d). Cows that did not conceive were included as a fourth days open category (not pregnant) in order to include the full database. Cows were also classified based on days to OLA (<21, ≥21 DIM; Chen J., 2015a). Weekly plasma concentrations of glucose, NEFA, BHB, IGF-1, insulin, and EB in week 1 till 7 postpartum. The natural logarithm of the plasma NEFA and BHB concentration were calculated to approximate normal distribution of these variables and were used in all statistical analyses. Subsequently, fixed effects in the model were either days open class or OLA class, and always transition treatment (0-d DP (LOW), 0-d DP (STD), 30-d DP (STD)) and parity class (2 or ≥3) and time (week relative to calving). Cow was considered as the repeated subject. Model assumptions were evaluated by examining the distribution of residuals. Values are presented as the least square mean with their standard errors of the mean. Differences were regarded as significant if *p* < 0.05, and trends were discussed if *p* < 0.10. When present (*p* < 0.05), interactions were clarified in a figure.

## 3. Results

The actual dry period length (DP) of cows with 30-d DP (STD) was 30 ± 6 d, the DP of cows with 0-d DP (LOW) or 0-d DP (STD) was 0 d. Results on EB and metabolic status after different transition treatments in the current experiment were reported earlier [9]. In short, reducing the level of energy in early lactation for cows after 0-d DP reduced EB both from week 4 till 7 [9] and from week 8 till 44 [23] postpartum, compared with a standard energy level after 0-DP. Postpartum, EB of cows with a 30-DP was more negative than of cow with a 0-d DP, the more negative EB was reflected in a greater plasma NEFA and BHB concentration, and a lower plasma glucose, insulin and IGF-1 concentration during weeks 4 and 7, compared with cows with a 0-d DP.

### 3.1. Effect of Transition Treatments on Days Open and Ovarian Activity

The effect of transition treatments on days open was dependent on parity (Table 2; Figure 2a). For young cows (parity 2), there was no effect of different transition treatments on days open (99.9, 96.6 vs. 119.9 d for 0-d DP (LOW), 0-d DP (STD) vs. 30-d DP (STD); *p* = 0.13). Older cows (parity ≥ 3), however, had less days open with 0-d DP (LOW), compared with older cows with 0-d DP (STD) (105.2 vs. 182.3, 137.8 d for 0-d DP (LOW) vs. 0-d DP (STD), 30-d DP (STD); *p* < 0.01).

For young cows, there was no effect of different transition treatments on services per conception. For older cows, cows with 0-d DP (LOW) or 30-d DP (STD) had less services per conception compared with older cows with 0-d DP (STD) (2.68, 2.71 vs. 4.31 for 0-d DP (LOW), 30-d DP (STD) vs. 0-d DP (STD); *p* < 0.01, respectively) (Figure 2b).

Cows with 0-d DP (LOW) had the greatest percentage of cows pregnant within 100 DIM compared with cows with 0-d DP (STD) or 30-d DP (STD), but there was no effect on the percentage of cows pregnant within 44 weeks. Cows with 0-d DP (LOW) had less days to first OLA compared with cows with 30-d DP (STD) (*p* < 0.01). Within 100 DIM, cows with a 0-d DP had more ovarian cycles compared with cows with 30-d DP length (*p* < 0.05). Among these cycles, cows with 0-d DP (LOW) had a lower percentage of prolonged cycles compared with cows with 0-d DP (STD) or 30-d DP (STD). For the first ovarian cycle, cows with 0-d DP (LOW) had shorter luteal phase and longer follicular phase compared with cows in group 0-d DP (STD).

### 3.2. Relationships between Days Open and Ovarian Activity

Cows with short (<80 d) and medium (80–130 d) days open had less days from calving till first AI compared with cows with long days open (>130 d) (*p* < 0.01) (Table 3). Cows with short (<80 d) and medium (80–130 d) days open had less services per conception compared with cows with long days open (>130 d) (*p* < 0.01).

Relations between days open and OLA, percentage of normal, short and prolonged cycles and cycle length were depended on transition treatments. Cows with medium days open (80–130 d) with a 0-d DP had less days to OLA compared with cows with medium days open with 30-d DP (18.64 d vs. 18.10 d, 28.62 d for 0-d DP (LOW) vs. 0-d DP (STD), 30-d DP (STD) (*p* = 0.01)) (Figure A1a). In addition, cows with medium days open (80–130 d) and 0-d DP (LOW) had a greater percentage of normal regular cycles compared with cows with medium days open with 0-d DP (STD) (48.50 vs. 25.75%; *p* = 0.04) (Figure A1b). Cows with days open less than 130 d (short and medium) had a lower percentage of prolonged ovarian cycles compared with cows with long days open (>130 d) and cows that did not get pregnant at all (*p* < 0.01). Cows that did not get pregnant had lower percentage of short cycles compared with cows in other groups (*p* < 0.01).

For the first ovarian cycle postpartum, cows with short (<80 d) and medium (80–130 d) days open had shorter luteal phase and shorter cycle length, compared with cows with long days open (>130 d) (*p* = 0.01). There was a tendency that cows with short (<80 d) and medium (80–130 d) days open had shorter follicular phase compared with cows with long days open (>130 d) (*p* = 0.07).

### 3.3. Relationships between Onset of Luteal Activity and Metabolic Status

Cows with OLA less than 21 DIM had greater glucose, IGF-1 and insulin concentration, better EB and lower NEFA concentration compared with cows with OLA equal or greater than 21 DIM (*p* < 0.05) (Table 4). For cows with OLA at less than 21 DIM, cows with 0-d DP (LOW) or 0-d DP (STD) had greater insulin concentration than cows with 30-d DP (STD) (16.69, 18.93 vs. 11.81 µIU/mL for 0-d DP (LOW), 0-d DP (STD) vs. 30-d DP (STD); *p* < 0.01).

### 3.4. Relationships between Days Open and Metabolic Status

Relations of days open with plasma concentration of insulin and EB in the first 7 weeks of lactation were dependent on parity (Table 5). Young cows with long (>130 d) and short (<80 d) days open had a more positive EB compared with young cows with a medium days open (80–130 d) and cows that did not get pregnant at all (*p* = 0.01) (32.89, −15.69 vs. −60.47, −178.09 kJ/kg^0.75^·day for long, short vs. medium days open, not pregnant) (Figure 2c). There was a trend that young cows with long days open (>130 d) had a greater insulin concentration in week 1 till 7 of lactation compared with young cows that did not get pregnant at all (17.47 vs. 10.22 µIU/mL) (*p* = 0.07). Cows with short and medium days open (<130 d) had a higher plasma insulin concentration, compared with cows that did not get pregnant at all.

Although EB in the first 7 weeks of lactation was negatively related with days open, this relationship was not so clear when the EB in the first 14 weeks was evaluated in relation with days open class. There was a tendency for the most negative EB in the first 14 weeks for cows with medium days open, compared with cows with short and long days open, which is also illustrated by Figure 3.

## 4. Discussion

Reducing postpartum dietary energy level for older cows (parity ≥ 3) after no DP (0-d DP (LOW)) reduced days open with 77.1 and 32.5 days compared with older cows fed postpartum a standard dietary energy level after no DP (0-d DP (STD)) or after a short DP of 30 d (30-d DP (STD)). This reduction in days open was partly related with less days postpartum to OLA, partly to less services per conception, and possibly to less prolonged cycles for cows with 0-d DP (LOW), compared with cows with 0-d DP (STD) or cows with 30-d DP (STD). In addition, in earlier studies, omitting of the DP resulted in a reduced interval from calving to first ovulation and less days open compared with 28 or 56 d dry period [4,24]. Our results also show that cows with 0-d DP had less days to OLA and more cycle numbers within 100 DIM compared with cows with 30-d DP. Several studies have observed positive effects of an earlier first ovulation after calving on fertility in dairy cows [25,26,27]. Cows with early first ovulation had more ovulatory cycles before first service compared with cows with late ovulation [27]. Minimizing the interval to first ovulation provides ample time for multiple ovarian cycles prior to insemination, which in turn improves conception rate and reduces inseminations per pregnancy [28]. In our study, a dietary energy level was similar among transition treatments till 21 DIM. Part of the cows ovulated before the dietary energy level contrast was present (32/42, 28/43 and 19/43 for 0-d DP (LOW), 0-d (STD) and 30-d (STD). This implies that only for cows with a delayed first ovulation (>21 DIM) could the dietary energy level contrast have an effect. This resulted in the fact that, on average, cows with 0-d DP (LOW) had a 4.4 days earlier OLA than cows with 0-d DP (STD). Moreover, the older cows (parity ≥ 3) with 0-d DP (LOW) also had 1.6 fewer services per conception than cows with a 0-d DP (STD), which contributed more to a reduction in days open of older cows with 0-d DP (LOW). Gumen et al. [4] also found that the number of services per conception was lower for cows with no DP (1.75) than for cows with a standard DP (3.00), with cows with a short DP being intermediate (2.44).

No improvement in days open was found in young cows with different transition treatments in our study. Watters et al. [1] also reported that reduction in days open (20 d) and an increase in pregnancy rate per insemination after 34-d DP were only observed in older cows (parity ≥ 3) and not in young cows (cows going from first to second lactation) when compared with cows with 55-d DP. Smith and Wallace [29] reported that ovulation before 21 DIM was associated with reduced pregnancy rates, increased services per pregnancy, and a prolonged calving to pregnancy interval for multiparous, but not primiparous dairy cows. In our results, both younger and older cows had fewer days to OLA after no DP, compared with cows with a short DP, but only older cows had shorter days open. Differences in response of older vs. younger cows to shortening or omitting the DP to fertility measures could be due to the relative priority of young cows for growth, as observed in another study on shortening of the DP [1] and a study on improving metabolic status by a more glucogenic diet [23].

In the current study, cows with 0-d DP had more ovarian cycles within 100 DIM compared with cows with 30-d DP. In addition, cows with 0-d DP (LOW) had a lower incidence of prolonged cycles compared with cows with 0-d DP (STD) or cows with 30-d DP (STD), which can be related to shorter luteal phase length and follicular phase length. The prolonged luteal phase is one of the most common ovarian disturbances in dairy cows [30]. Studies reported that cows with prolonged luteal phase had a lower first AI conception rate, more services per conception, and more days open compared with cows with normal ovarian cycles [30,31,32]. The most important risk factors for developing prolonged luteal phases are puerperal problems, such as metritis and mastitis. The ratio of prostaglandin E2 (PGE2) to prostaglandin F2α (PGF2α) decides the fate of the corpus luteum, with persistence if PGE2 dominates or luteolysis if PGF2α dominates [33]. Lipopolysaccharide stimulates the secretion of prostaglandins and particularly PGE2 [34]. Metritis and mastitis could compromise release of prostaglandin F2α (PGF2α) [30,35,36] by increasing lipopolysaccharide in postpartum cows [37], thereby delaying luteolysis, which results in a prolonged luteal phase [38]. In our study, however, no significant differences were found in the incidence of metritis and mastitis between the different transition treatments, the incidence of metritis in two groups were 7.1% and 4.7%, though the incidence of mastitis in transition treatment 0-d DP (LOW) was 31%, which was numerically lower than 40% in transition treatment 0-d DP (STD). A high milk yield is also one of the major risk factors for a prolonged luteal phase in high-producing dairy cows [39]. In addition, in the current study, the milk yield of cows with 30-d DP (STD) was greater than that of cows with 0-d DP (LOW) or 0-d DP (STD) (30.3 vs. 24.2, 24.7 kg/d) (*p* < 0.01).

The second objective of the current study was to relate days open of dairy cows after different transition treatments to underlying reproductive measures. Short (<80 d) and medium (80–130 d) days open was related to a short interval form calving to first AI, compared with long days open. This is in accordance with Harrison [40], who observed that average days open was positively related (r^2^ = 0.99) to the interval to first AI in a field study in 12 commercial dairy herds in Michigan. In that study, the first insemination was directly decided by the time of the first ovulation after calving. Additionally, prolonged days open was related with a high number of services per conception [41]. In our study, short (<80 d) days open was related to less days to OLA, which can also partly reveal the occurrence of delayed resumption of ovarian cyclicity [30,35]. Gautam et al. [42] reported that delayed resumption of ovarian activity adversely influenced the hazard of pregnancy, which was partly driven by a delay in first AI as well as by a substantial reduction in the first AI conception rate. In our study, more days open was not only related to delayed OLA, but also to an increased number of services per pregnancy.

The third objective of the current study was to relate days open and OLA of cows after different transition treatments to their metabolic status in early lactation. Independent of transition treatment with OLA at less than 21 DIM had a better EB during weeks 1 to 7 and the concentration of IGF-1 and insulin were greater compared with cows with OLA equal or greater than 21 DIM. The more negative EB in cows with OLA equal or greater than 21 DIM is possibly related with a compromised ovarian follicular development by suppressed plasma IGF-1 concentration and pituitary luteinizing hormone (LH) pulsatility [43]. This negatively impacts reproduction as IGF-1 is unable to synergise with the gonadotrophins on ovarian cells preventing the dominant follicle from ovulating [44] and delaying the resumption of cyclicity [45], at last leading to a prolonged interval from calving to first ovulation [30]. We also noticed that cows with OLA equal or greater than 21 DIM had a lower concentration of glucose and tended to have a greater NEFA concentration compared with cows with OLA at less than 21 DIM. Greater concentrations of circulating NEFAs were associated with lower follicular estradiol concentrations, impairing ovulation of the dominant follicle [46]. Prolonged intervals from calving to first ovulation have been related to uterine infection [35], mastitis [47,48], and lameness [48]. Like discussed above, also, in the current study, cows with more prolonged cycles (0-d DP (STD) had the numerically greatest incidence of mastitis in early lactation.

In the current study, cows with long days open (>130 d) or cows that did not get pregnant at all had greater plasma concentration of NEFA as well as a more severe negative EB than cows with short (<80 d) days open. Multiple studies have reported a negative relationship of NEFA with reproduction [49,50]. Increased NEFA concentrations during the transition period were associated with decreased pregnancy rate at first AI [46] or at 70 d after the voluntary waiting period [51], whereas another study in 60 free stall herds found that high circulating NEFA was associated with a reduced 21-d pregnancy rate after a voluntary waiting period [52]. All of these studies found positive relations between plasma NEFA and days to OLA and days open, as reported in the current study. Greater concentration of NEFA during a period of negative EB may prevent follicle development, interrupt the complicated endocrine system, and advance the formation of ovarian cysts [53]. In contrast, a low plasma NEFA concentration, combined with a positive EB, was maybe related to an earlier postovulatory increase of progesterone, a greater follicle development, and a better resumption of ovarian activity, resulting in fewer days open. In addition, long days open in young cows could be related with a too positive EB and high insulin concentration, while the cows with short days open had the medium insulin concentration and EB. As shown by Armstrong [13], high energy intake leading to higher insulin concentrations increased the growth rate of the dominant follicle but impaired oocyte quality. Energy balance was positively correlated with the number of large follicles in ovary, and negatively correlated with the numbers of small and medium follicles [54]. However, cows with medium-sized follicles (between 14.5 and 17.5 mm) had a greater pregnancy rate than cows with follicles of other sizes [55].

Remarkably, multiparous cows after 0-d DP and fed postpartum a standard energy level did not have a reduction in days open, as we saw for cows after 0-d DP and fed postpartum a low energy level, compared with cows which were practically overfed like cows on 0-d (STD). Uterine recovery after parturition and at the same time producing milk to feed the new-born, and remain a healthy and functional dairy cow, can be considered highly energy-demanding processes. Cows fed postpartum a standard energy level after 0-d DP, however, were characterized by more days open, lower pregnancy rates within 100 DIM, more services per conception and more prolonged cycles, compared with cows fed postpartum a low energy level after 0-d DP. This is in line with Watters et al. [1], that both young and old cows with a short DP (34 d) and fed a low postpartum energy level had fewer days to first ovulation, compared with cows with a traditional DP (55 d). Still, in our study, only the older cows had earlier time to pregnancy when postpartum dietary energy level was reduced after 0-d DP. It is unknown why cows with 0-d DP (STD) treatment had a lower fertility (more days open, lower pregnancy rates within 100 DIM) compared with cows with 0-d DP (LOW) treatment. It can be speculated, however, that the EB is possibly too positive in early lactation in this group (−2 vs. 55 kJ/kg^0.75^ day for 0-d DP (LOW) vs. 0-d DP (STD), respectively) [9]. As discussed above, too high plasma insulin concentration had negative effects on oocyte maturation in vitro [19] and insulin stimulating diets are possibly beneficial to establish ovarian activity postpartum, but not to establish pregnancy [20]. Diets designed to increase plasma insulin concentration had negative effects on blastocyst rate in heifers [56,57] and in lactating dairy cows [58]. Moreover, this implies that not only a severe NEB is detrimental for reproductive performance, but also a positive EB with elevated insulin levels in plasma can have negative consequences for reproduction in dairy cows. This is in line with our findings, where, on the one hand, early OLA was related to a greater plasma insulin concentration, while, on the other hand, days open was not related to plasma insulin concentration in early lactation. Moreover, a nonlinear relationship between EB and days open would clarify why both the treatment group with the best EB in early lactation (0-d DP (STD)) and the treatment group with the most negative EB (30-d DP (STD)) have similar days open which was longer that the treatment group with an intermediate EB (0-d DP (LOW)). It can be speculated that cows with 0-d DP (LOW) had a more optimal EB than cows with a 0-d DP (STD) or 30-d DP (STD) to support fertility.

## 5. Conclusions

Reducing postpartum dietary energy level for older cows (parity ≥ 3) after no DP (0-d DP (LOW)) improved fertility by reducing the interval from calving to OLA, reducing services per conception and consequently reducing days open compared with a standard dietary energy level after no DP (0-d DP (STD)) or after 30 d (30-d DP (STD)). Less days to OLA (<21 d) was associated with a better metabolic status, indicated by a greater concentration of glucose, IGF-1, and insulin and a lower concentration of NEFA and BHBA during weeks 1 through 7 postpartum. A low number of days open (< 80-d) was associated with less days to OLA, less services per conception, better EB in week 1 till 7 of lactation and better metabolic status. Energy balance in week 1 till 14 of lactation, however, was not linearly related with days open. This might indicate that cows with an intermediate EB in week 1 till 14 of lactation (0-d DP (LOW)) had a more optimal EB to support fertility than cows with a positive EB (0-d DP (STD)) or cows with most negative EB (30-d DP (STD)) in week 1 till 14 of lactation.

## Figures and Tables

**Figure 1 animals-10-01100-f001:**
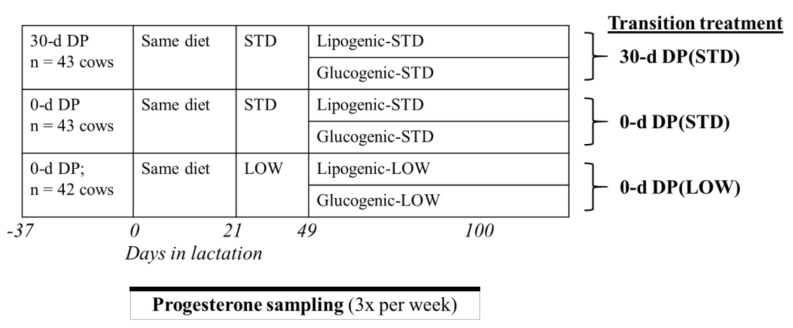
Overview of experimental design and sampling protocol for cows with different transition treatments. 30-d DP = 30 days dry period; 0-d DP = 0 day dry period; STD = standard energy level; LOW = low energy level.

**Figure 2 animals-10-01100-f002:**
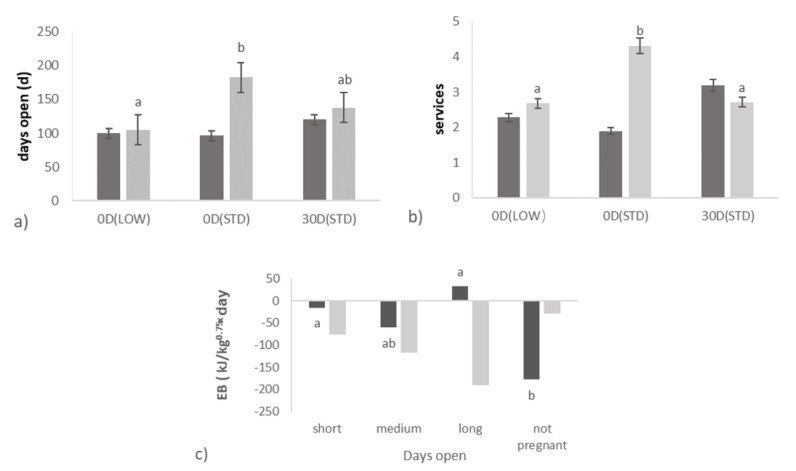
The effect of transition treatments (0-d DP (LOW), 0-d DP (STD), 30-d DP (STD) for cows of different parity classes (parity = 2 in dark grey or parity ≥ 3 in light grey) on days open (**a**), services per pregnancy (**b**); The relation of days open class (short:<80 d, medium: 80–130 d, long:>130 d and not pregnant) for cows of different parity classes (parity = 2 in dark grey or parity ≥ 3 in light grey) with energy balance (EB) in week 1 till 7 of lactation (**c**); ^a,b^ Values within parity class in the same row with different superscripts differ (*p* < 0.05).

**Figure 3 animals-10-01100-f003:**
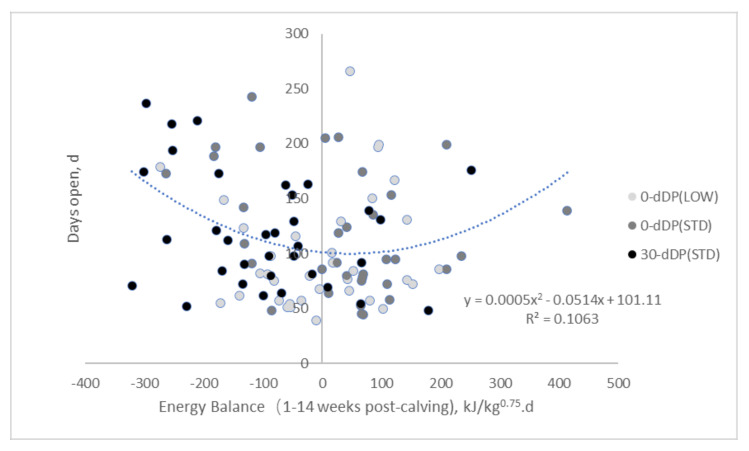
Energy balance for cows with different transition treatments in 1–14 weeks post-calving. Transition treatments were: 0-d dry fed low energy level postpartum (0-d (LOW)); 0-d dry fed a standard energy level postpartum (0-d (STD)); 30-d dry fed a standard energy level postpartum (30-d (STD)).

**Table 1 animals-10-01100-t001:** Distribution of cows with 0 (0-d DP) or 30 days (30-d DP) of dry period and fed a low (LOW) or standard (STD) dietary energy level.

Cows, n	0-d DP (LOW)	0-d DP (STD)	30-d DP (STD)	Total
Cows in experiment	42	43	43	128
Cows with OLA activity within 100 DIM ^1^	42	42	43	127
Cows with complete 1st ovarian cycle	41	39	39	119
Cows with complete 2nd ovarian cycle	32	26	29	87

^1^ DIM = Days in milk.

**Table 2 animals-10-01100-t002:** Reproduction measures and incidence of normal and abnormal ovarian cyclicity within 100 days in milk (DIM) of cows with different transitiontreatments ^1^.

			Transition Treatments				SEM	*p*-Value ^2^		
	0-d DP (LOW)		0-d DP (STD)		30-d DP (STD)			TT	P	TT × P
Parity	2	≥3	2	≥3	2	≥3				
Cows (n)	21	21	22	21	21	22				
Days open (days)	99.9 ^a^	110.7 ^a^	96.6 ^a^	159.3 ^b^	114.9 ^a,b^	127.5 ^a,b^	15.0	<0.01	<0.01	<0.01
Calving to first AI ^3^ (days)	73.5	66.9	66.1	71.9	67.7	69.0	5.0	0.97	0.48	0.50
Services per conception	2.3 ^b^	2.7 ^b^	2.1 ^b^	4.3 ^a^	2.9 ^a,b^	2.7 ^b^	0.4	0.27	0.02	<0.01
Pregnant within 44 weeks (%)	95.2	95.2	95.4	87.1	85.2	95.5	7.1	0.96	0.90	0.99
Pregnant within 100 DIM (%)	66.7 ^a^	57.1 ^a,b^	59.1 ^a,b^	23.8 ^b^	42.9 ^a,b^	22.7 ^b^	10.4	0.02	0.04	0.63
Days to 1st OLA ^4^	17.0 ^a^	21.9 ^a,b^	20.2 ^a,b^	27.6 ^b^	26.3 ^a,b^	28.3 ^b^	2.6	<0.01	<0.01	0.31
Cycle number per cow within 100 DIM	2.1 ^a,b^	2.0 ^a,b^	2.2 ^a^	1.8 ^a,b^	1.9 ^a,b^	1.7 ^b^	0.2	<0.01	<0.01	0.07
Normal cycles (per cow within 100 DIM) (%)	48.6	39.6	41.7	39.7	42.0	42.1	8.4	0.48	0.44	0.29
Short cycles (per cow within 100 DIM) (%)	16.5 ^a,b^	18.1 ^a,b^	18.5 ^a,b^	5.9 ^a^	23.2 ^b^	7.0 ^a,b^	5.6	0.06	<0.01	<0.01
Prolonged cycles (per cow within 100 DIM) (%)	34.9 ^a^	42.4 0 ^a,b^	40.9 ^a,b^	54.4 ^b^	34.8 ^a,b^	50.9 ^a,b^	8.6	0.02	0.09	0.45
1st ovarian cycle postpartum (days)										
Luteal phase length (days)	18.0 ^a,b^	18.9 ^a,b^	21.0 ^a,b^	21.8 ^a^	13.9 ^b^	17.9 ^a,b^	2.7	<0.01	0.10	0.15
Follicular phase length (days)	6.9 ^a,b^	10.4 ^a^	7.4 ^a,b^	6.5 ^b^	8.2 ^a,b^	9.4 ^a,b^	1.3	<0.01	<0.01	<0.01
Cycle length (days)	24.9	28.3	28.4	28.3	22.2	27.3	2.9	0.02	<0.01	0.09

^a,b^ Values in the same row with different superscripts differ (*p* < 0.05) ^1^ Transition treatments were: 0-d dry fed low energy level postpartum (0-d (LOW)); 0-d dry fed a standard energy level postpartum (0-d (STD)); 30-d dry fed a standard energy level postpartum (30-d (STD)); ^2^ TT = Transition treatment; *p* = Parity. ^3^ AI = Artificial Insemination; ^4^ OLA = Onset of luteal activity.

**Table 3 animals-10-01100-t003:** Relationship between days open and characteristics of ovarian cycles of dairy cows with different transition treatments ^1^.

Variable	Days Open				SEM	Transition Treatments			SEM	*p*-Value ^2^			
	Short (<80 d)	Mid (80–130 d)	Long (>130 d)	Not Pregnant ^3^		0-d DP (LOW)	0-d DP (STD)	30-d DP (STD)		Days Open	P	TT	TT × Days Open
Cows, n	37	37	37	17		42	43	43					
Calving to first AI ^4^ (days)	60.42 ^a^	68.85 ^a^	80.16 ^b^	68.07 ^a^	5.55	71.02	67.44	69.66	6.03	<0.01	0.48	0.97	0.83
Services per pregnancy	1.44 ^a^	2.71 ^b^	4.09 ^c^		0.36	2.36	3.31	2.57	0.16	<0.01	0.35	0.79	0.79
Days to first OLA ^5^	24.73 ^b^	22.20 ^a^	23.76 ^c^	24.68 ^c^	1.36	19.26	25.53	26.73	2.84	<0.01	<0.01	<0.01	<0.01
Cycle number (per cow within 100 DIM)	1.62 ^a^	2.49 ^c^	2.00 ^b^	1.74 ^a^	0.06	2.32	1.77	1.80	0.20	<0.01	<0.01	<0.01	0.22
Normal cycles (per cow within 100 DIM) (%)	47.96 ^a^	43.98 ^a^	30.47 ^b^	38.83 ^a b^	4.66	42.28	40.02	38.62	6.41	<0.01	0.41	0.82	<0.01
Short cycles (per cow within 100 DIM) (%)	18.01 ^a^	14.93 ^a^	13.74 ^a^	6.76 ^b^	3.06	14.40	10.65	15.04	6.45	<0.01	<0.01	0.19	<0.01
Prolonged cycles (per cow within 100 DIM) (%)	35.03 ^a^	40.92 ^a^	55.31 ^b^	52.90 ^b^	4.88	42.17	49.26	46.68	9.75	<0.01	0.10	0.66	<0.01
1st ovarian cycle postpartum (days)													
Luteal phase length	17.94 ^a^	18.22 ^a^	22.03 ^b^	18.29 ^b^	0.74	20.18	21.01	16.16	2.92	<0.01	0.82	<0.01	0.05
Follicular phase length	7.02	7.33	9.72	7.73	0.77	8.83	6.59	8.43	1.38	0.07	0.01	<0.01	0.31
Cycle length	25.00 ^a^	25.28 ^a^	31.89 ^b^	25.56 ^a^	1.59	28.58	27.56	24.66	3.19	<0.01	0.30	<0.01	<0.01

**^a,b^ Values in the same row with different superscripts differ (*p* < 0.05**). ^1^ Transition treatments were: 0-d dry fed low energy level postpartum (0-d (LOW)); 0-d dry fed a standard energy level postpartum (0-d (STD)); 30-d dry fed a standard energy level postpartum (30-d (STD)); ^2^ TT = Transition treatment; *p* = Parity. ^3^ Cows did not get pregnant through the lactation. ^4^ AI = Artificial Insemination; ^5^ OLA = Onset of luteal activity. None of the variables had an interaction between Days open class and parity class, treatments and parity class.

**Table 4 animals-10-01100-t004:** Relationship between days to onset of luteal activity (OLA) and postpartum ^1^ plasma metabolites and metabolic hormones of cows after different transition treatments ^2^.

	Days to OLA		SEM	Transition Treatments			SEM	*p*-Value ^3^						
	<21 d	≥21 d		0-d DP (LOW)	0-d DP (STD)	30-d DP (STD)		OLA	TT	P	W	OLA × TT	OLA × P	OLA × W
Cows, *n*	66	53											
Glucose (mmol/L) ^4^	3.92 ^a^	3.71 ^b^	0.05	3.89	3.90	3.66	0.04	<0.01	<0.01	<0.01	<0.01	0.21	0.15	0.65
NEFA (mmol/L) ^4^	0.11 ^a^	0.20 ^b^		0.12	0.12	0.23		<0.01	<0.01	<0.01	<0.01	0.47	0.01	0.24
	(0.10–0.13)	(0.17–0.22)		(0.11–0.14)	(0.10–0.14)	(0.20–0.26)								
BHB (mmol/L )^4^	0.63 ^a^	0.73 ^b^		0.66	0.64	0.74		<0.01	0.02	<0.01	0.01	0.01	0.07	0.36
	(0.59–0.67)	(0.68–0.79)		(0.60–0.72)	(0.59–0.69)	(0.69–0.80)								
IGF-1 (ng/mL) ^4^	129.11 ^a^	95.31 ^b^	6.48	122.63	121.82	92.17	5.39	<0.01	<0.01	<0.01	<0.01	0.29	0.43	0.01
Insulin (µIU/mL) ^4^	15.86 ^a^	11.41 ^b^	0.82	14.58	15.51	10.81	0.68	<0.01	<0.01	0.86	<0.01	0.10	0.71	0.08
EB (kJ/kg^0.75^ ·day) (week 1 till 7) ^5^	−15.52 ^a^	−153.05 ^b^	26.61	−55.59	−18.33	−178.94	22.11	<0.01	<0.01	0.06	<0.01	0.22	<0.01	0.19

^a,b^ Values with different superscripts differ (*p* < 0.05). ^1^ Postpartum = weeks 1 to 7 after calving. ^2^ Transition treatments were: 0-d dry fed low energy level postpartum (0-d(LOW)); 0-d dry fed a standard energy level postpartum (0-d (STD)); 30-d dry fed a standard energy level postpartum (30-d (STD)). ^3^ TT= Transition treatment; *p* = Parity; W = Week relative to calving. ^4^ Concentration in plasma was measured weekly between weeks 1–7 post-calving. Non-esterified fatty acids (NEFA) and β-hydroxybutyrate (BHB) were log transformed for analysis, but are shown as actual values with confidence interval. ^5^ EB = Energy balance. None of the variables had an interaction between treatments and parity class.

**Table 5 animals-10-01100-t005:** Relationship between days open classes and postpartum ^1^ plasma metabolites and metabolic hormones of cows after different transition treatments ^2^.

	Days Open				SEM	Transition Treatments			SEM	*p*-Value ^2^						
	<80 d	80-130 d	>130 d	Not Pregnant ^5^		0-d DP (LOW)	0-d DP (STD)	30-d DP (STD)		Days Open	TT	P	W	Days Open × P	Days Open × TT	Days Open × W
	34	34	37	13												
Glucose (mmol/L) ^3^	3.86	3.81	3.81	3.71	0.62	3.87	3.91	3.64	0.05	0.89	<0.01	0.04	<0.01	0.73	0.88	0.28
NEFA (mmol/L) ^3^	0.12	0.15	0.15	0.16		0.13	0.10	0.22		0.12	<0.01	<0.01	<0.01	<0.01	0.44	0.53
	(0.09–0.14)	(0.13–0.18)	(0.12–0.18)	(0.12–0.20)		(0.11–0.16)	(0.08–0.12)	(0.19–0.26)								
BHB (mmol/L) ^3^	0.66	0.67	0.73	0.66		0.69	0.62	0.73		0.53	0.03	0.12	<0.01	0.43	0.85	0.53
	(0.59–0.73)	(0.61–0.74)	(0.66–0.80)	(0.58–0.76)		(0.62–0.76)	(0.56–0.68)	(0.67–0.80)								
IGF-1 (ng/mL) ^3^	122.26	116.49	111.48	105.99	7.24	122.04	128.34	91.79	6.61	0.72	0.73	0.37	<0.01	0.42	0.87	0.10
Insulin (µIU/mL) ^3^	14.88 ^a^	14.07 ^a^	13.75 ^a b^	12.12 ^b^	0.98	14.04	16.42	10.65	0.78	<0.01	<0.01	0.42	<0.01	<0.01	0.92	0.57
EB (kJ/kg^0.75^ ·day) (week 1 till 7) ^4^	−48.34 ^a^	93.09^b^	92.98^b^	−118.71 ^b^	31.64	−87.88	2.54	−179.51	38.84	<0.01	0.09	0.90	<0.01	<0.01	0.58	0.21
EB (kJ/kg^0.75^·day) (week 1 till 14)	−15.86	−26.47	−10.98	−33.50	21.53	−23.32	49.36	−91.15	20.60	0.05	0.02	0.02	0.36	0.05	0.72	0.37

^a,b^ Values with different superscripts differ (*p* < 0.05). ^1^ Postpartum = weeks 1 to 7 after calving, unless otherwise stated; ^2^ Transition treatments were: 0-d dry fed low energy level postpartum (0-d(LOW)); 0-d dry fed a standard energy level postpartum (0-d (STD)); 30-d dry fed a standard energy level postpartum (30-d (STD)); ^3^ TT = Transition treatments; *p* = Parity; W = Week relative to calving. ^4^ Concentration in plasma was measured weekly between weeks 1–7 post-calving. NEFA and BHB were log transformed for analysis, but are shown as actual values with confidence interval. ^5^ EB = Energy balance. ^6^ Cows did not get pregnant through the lactation. None of the variables had an interaction between treatments and parity class.

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
