# Peer review of "Consequences of Transition Treatments on Fertility and Associated Metabolic Status for Dairy Cows in Early Lactation"

_animals, 2020, doi:10.3390/ani10061100_

Round 1
Reviewer 1 Report
The manuscript has reported the results of studies which showed reducing postpartum dietary energy level for older cows (parity ≥ 3) after no DP (0-d DP(LOW)) improved fertility by reducing the interval from calving to OLA, reducing services per conception and consequently reducing days open. These effects were associated with better metabolic profile in these cows. The manuscript is well-written and contains extensive statistical analysis of the data. There are several points which I this needs to be addressed by the authors before acceptance of the manuscript for publication.
- I suggest to include a diagram to explain the experimental design.
- The experimental design is confusing. In lines 117-118 says the cows were blocked with 6 cows in each block. How many blocks were designed? The information needs to be clearer.
- Data presentation in the table does not match with the explanation of the results in the text. In fact, the text explains the hidden data rather than explaining the tables. I suggest arranging presentation of the tables based on the experimental design as a nested data. For example, in table 2, data for parity be presented directly under the transition treatments (not in parallel). So, under 0-d DP(LOW) you will have two columns for one for parity 2, and one 3 or over, and then the number of animals in each parity followed by all the other parameters are written in different rows for each parity. Similar changes shall be made to 0-d DP(STD) and 30-d DP(STD) transition treatments. This will show clearly how many cows you had from each parity in each transition treatment group, and whether the differences were significant or not. Similar changes shall be made to tables 3-5.
- Table 3, Days to first OLA values are 19.26 for 0-d DP(LOW) group and 25.53 for 0-d DP(STD). According to the experimental design, these cows received a similar diet until 3 weeks after parturition. Can the authors justify such differences?
- Lines 146-150. Indicate inter and intra assay variations.
- Finally, the manuscript lacks information about the physiological basis that can explains the favourable changes in the reproductive parameters of cows in 0-1 DP(LOW). Uterine recovery after parturition and at the same time producing milk to feed the new-born, and remain a healthy and functional dairy cow, are highly energy-demanding processes in dairy cows.
Reviewer 2 Report
Great paper, really interesting results. The only problem with this paper is the writing of the first objective (lines 93-97) needs to be rewritten. It is disorganized and very hard to understand.
Also, sometimes the writers use 'no dry' and sometimes 0 d DP. Please use or the other throughout the paper.
Phrases are out of order from lines 89- 93 which also makes reading more difficult.
Suggestions:
line 89 remove 'like explained above' and insert Earlier,
then remove 'earlier for the current experient'
line 92 remove 'on days open' and insert after energy level on line 93
line 93 remove 'More specifically'
line 109 insert 'barn' after free stall
